# Impact of Chemotherapy Regimens on Body Composition of Breast Cancer Women: A Multicenter Study across Four Brazilian Regions

**DOI:** 10.3390/nu15071689

**Published:** 2023-03-30

**Authors:** Jordana Carolina Marques Godinho-Mota, Larissa Vaz-Gonçalves, Isis Danyelle Dias Custódio, Jaqueline Schroeder de Souza, João Felipe Mota, Maria Cristina Gonzalez, Priscylla Rodrigues Vilella, Karine Anusca Martins, Yara Cristina de Paiva Maia, Sara Maria Moreira Lima Verde, Aline Porciúncula Frenzel, Patricia Faria Di Pietro, Eduarda da Costa Marinho, Ruffo Freitas-Junior

**Affiliations:** 1Csentro Avançado de Diagnóstico da Mama, Clinics Hospital, Federal University of Goias, Goiania 74175-120, Brazil; larivaznutri@gmail.com (L.V.-G.); ruffojr@terra.com.br (R.F.-J.); 2Nutrition Faculty, Federal University of Goias, Goiania 74690-900, Brazil; joao_mota@ufg.br (J.F.M.); priscyllarvilella@gmail.com (P.R.V.); karine_anusca@ufg.br (K.A.M.); 3Faculty of Medicine, School of Public Health, Queensland University, Brisbane 4006, Australia; 4Faculty of Medicine, Federal University of Uberlandia, Uberlandia 38408-100, Brazil; isisdanyelle@yahoo.com.br (I.D.D.C.); yara.maia@ufu.br (Y.C.d.P.M.); eduarda@hospitaldocancer.org.br (E.d.C.M.); 5Post Graduate Program in Nutrition, Federal University of Santa Catarina, Florianopolis 88040-900, Brazil; jaqueline.schroeder@posgrad.ufsc.br (J.S.d.S.); patricia.di.pietro@ufsc.br (P.F.D.P.); 6Post-Graduate Program in Health and Behavior, Catholic University of Pelotas, Pelotas 96015-560, Brazil; cristina.gonzalez@ucpel.edu.br (M.C.G.); aline.frenzel@hotmail.com (A.P.F.); 7Post-Graduate Program in Nutrition and Health, State University of Ceara, Fortaleza 60714-903, Brazil; sara.maria@uece.br

**Keywords:** breast cancer, adjuvant treatment, body mass index, fat mass index, fat-free mass index

## Abstract

This study aimed to investigate the effect of chemotherapy (CT) and its different types of regimens on the anthropometry and body composition of women with breast cancer. Three-hundred-and-four women with breast cancer were enrolled in this multicenter study. The participants were evaluated before the infusion of the first cycle of CT (pre-CT), and until two weeks after CT completion (post-CT), regarding body weight, body mass index (BMI); waist circumference (WC); waist-to-height ratio (WHtR); conicity index (C-index); fat mass index (FMI); and fat-free mass index (FFMI). CT regimens were classified as anthracycline-based (AC—doxorubicin or epirubicin); anthracyclines and taxane (ACT); cyclophosphamide, methotrexate, and 5-fluorouracil (CMF); or isolated taxanes (paclitaxel or docetaxel). Women significantly increased BMI and FMI post-CT (*p* < 0.001 and *p* = 0.007, respectively). The ACT regimen increased FMI (*p* < 0.001), while FFMI increased after AC (*p* = 0.007). It is concluded that the CT negatively impacted body composition and the type of regime had a strong influence. The ACT regimen promoted an increase in FMI compared to other regimens, and the AC increased FFMI. These findings reinforce the importance of nutritional monitoring of breast cancer patients throughout the entire CT treatment.

## 1. Introduction

Brazil presents a multidimensional socioeconomic context throughout its five regions. Socioeconomic status (SES)—comprised of factors such as income, education, poverty, and wealth—can be different through the regions; however, the overall combination of these factors represents the whole Brazilian nation. The SES can be linked to the prevalence in rates of and mortality from chronic diseases [1]. For instance, breast cancer is the most common cancer among women worldwide, excluding non-melanoma cancer. An incidence of 43.7 cases per 100,000 females was described in Brazil, representing 29.7% of all cancer types [2]. On the other hand, in historical analysis, mortality rates have been decreasing since the 1990s. For the year 2022, this rate was 11.84/100,000, being higher in the South and Southeast [2]. This improvement in survival rates is due to earlier detection and improved treatments [2,3].

Breast cancer treatments can be classified into two modalities: local (e.g., surgery and radiotherapy), or systemic (e.g., chemotherapy—CT, endocrine and biological therapy). The adequate treatment is chosen based on several factors: for example, molecular subtypes and stages of breast cancer, personal preferences, menopause status, and women’s overall health [4]. CT is the most commonly used treatment for breast cancer, being effective in improving prognosis, increasing overall and disease-free survival, and reducing the risk of recurrences [5]. In the adjuvant setting, cytotoxic chemotherapeutic agents are generally used in combination. The regimens include sequential cyclophosphamide, methotrexate, and 5-fluorouracil (CMF); anthracycline agents (AC), in the second generation; and a combination of anthracyclines and taxanes (ACT) for the third generation [6,7]. 

Although studies have reported the effectiveness of CT in breast cancer treatment [6], CT could promote short- and long-term side effects [7]. In this context, evidence has shown that CT negatively affects anthropometric and body composition parameters [8,9,10]. For instance, CT can increase body weight, body mass index (BMI), and central adiposity (waist circumference and fat mass) and reduce lean mass [8,11,12]. Furthermore, these changes can lead to a poor prognosis [13], presenting a higher risk of toxicity [14] and breast cancer recurrence [15,16]. Non-communicable chronic diseases (NCD) (e.g., type 2 diabetes and cardiovascular diseases), reduced life expectancy, and a lower quality of life can also be observed as a consequence of those changes [17,18]. 

The regimen (combination of the drugs) and type of treatment (neoadjuvant or adjuvant) used are important considerations that might be directly related to those changes among patients with breast cancer undergoing CT [10]. Changes in anthropometric and body composition parameters during and after CT have been described previously [8,9,10,11,12]. A meta-analysis reported a positive association between the type and regimen of CT and body weight [10]; however, this association has not been deeply explored—although the possible relationship between these variables has already been mentioned in the literature [19,20]. To the best of our knowledge, no studies have investigated the effect of CT on anthropometric and body composition parameters considering a representative sample of Brazilian women. We hypothesize that throughout CT, women with breast cancer will have higher BMI (weight gain); fat mass index (FMI); central adiposity; and a lower fat-free mass index (FFMI), which may differ between CT regimens. Therefore, the aim of this study was to investigate the effect of CT and its different types of regimens on the anthropometric and body composition parameters of breast cancer women patients. 

## 2. Materials and Methods

### 2.1. Study Design and Participants

This is a retrospective secondary analysis involving six states of four regions of Brazil, where data were prospectively collected (Figure 1). Using a multicentric analysis [21] we combined cohort studies of Central-West (Goiás, GO); Southeast (São Paulo, SP, and Minas Gerais, MG states); Northeast (Ceará, CE); and South (Rio Grande do Sul, RS, and Santa Catarina, SC). The study settings were the Clinical Hospital of the Federal University of Goiás (GO); the A. C. Camargo Cancer Centre Hospital (SP); the Clinical Hospital of the Federal University of Uberlandia (MG); Fortaleza General Hospital and the Integrated Regional Oncology Centre (CE) of the University Hospital of the Federal University of Pelota (RS); and the Carmela Dutra Maternity Hospital and Oncology Research Centre (SC). Each center had its own research protocol and used a standardized protocol for the recruitment of participants, collection of questionnaire data, and anthropometric and body composition measurements. As the study aims were similar in all centers, the research teams have harmonized the data for this study proposal.

The inclusion criteria were women with primary breast cancer undergoing CT without previous treatment for breast cancer or other cancer. All women aged 18 years or over, with clinical stages I to III, presenting available information related to CT treatment, tumor characteristics, anthropometric, and body composition data were included. Exclusion criteria were patients with mobility difficulties and any cognitive or psychiatric impairment that prevented understanding and data collection were also excluded.

### 2.2. Ethics Statement

This study was approved by the Research Ethics Committee from the Clinical Hospital of the Federal University of Goiás (n.3.858.331/2020). Written informed consent was provided by all participants after all risks, discomforts, and benefits involved in the study were reviewed in each region.

### 2.3. Data Collection

Data were collected from 2004 to 2018 following the study protocol in each region. The follow-up time was based on the CT regimen, presenting a range from four to six months. Trained researchers conducted a personal interview at the diagnosis or before the infusion of the first cycle (pre-CT—T0) and after the last CT cycle until two weeks after CT completion (post-CT—T1).

### 2.4. Socioeconomic Status and Tumor-Related Characteristics

Age (full years), self-reported skin color, marital status (with or without a partner), education level attained, family income/month, and menopausal status were collected during a personal interview in T0.

Morphological tumor types (ductal, lobular, mucinous); degree of differentiation, clinical stage (tumor node and metastasis—TNM); molecular subtypes (luminal, human epidermal growth factor receptor (HER2), and triple-negative); type of surgery (mastectomy/breast-conserving surgery); and CT regimen were obtained from the physician in charge or from medical records. CT regimens were classified as anthracycline-based; combined cyclophosphamide with doxorubicin or epirubicin (AC); anthracyclines plus taxanes (ACT); cyclophosphamide, methotrexate, fluorouracil (CMF); and isolated taxanes (PD—either paclitaxel or docetaxel).

### 2.5. Behavioral Variables 

The consumption of alcoholic beverages was calculated in grams per day according to frequency, quantity, and type of drink mentioned as habitual [22], and classified into ≥10 (risk for breast cancer) or <10 g/day alcohol intake (non-risk) [23]. Smoking status was determined based on current smokers (current smokers or quit smoking less than a year ago) or ex-smokers (stopped smoking more than 1 year ago) and non-smokers, based on a previous study [22]. Physical activity (PA) was assessed using the International self-administered Physical Activity Questionnaire short form (IPAQ-SF) [24], and volunteers were classified as active (≥150 min of moderate PA/week) or inactive (<150 min of moderate PA/week) [25]. These dates were collected in T0 for characterizing the sample.

### 2.6. Anthropometry and Body Composition

Anthropometric measurements were conducted according to Habicht procedures [26] and collected at T0 and T1. Height was evaluated by a vertical stadiometer with an accuracy of 0.1 cm (Model P150-C, Lider R, São Paulo, Brazil) and body weight was assessed by a digital scale accurate to 0.1 kg and with a capacity of 150 kg (Lider R or Filizola™, São Paulo, Brazil). BMI was calculated as a ratio between weight (kg) and height (m)^2^ using age-specific cut-off values for adults (normal weight: ≥18.5–24.9 kg/m^2^ and overweight: ≥25 kg/m^2^) [27]; and older adults (normal weight: ≥22–27 kg/m^2^ and overweight:>27 kg/m^2^) [28]. 

Waist circumference (WC) was measured at the midpoint between the lowest rib and the iliac crest using an inelastic measuring tape of 1 mm precision, and women were classified as having a low (<80 cm), high (≥80 cm), and very high (≥88 cm) risk for metabolic complications, according to the World Health Organization [27]. The waist-to-height ratio (WHtR) was calculated as a ratio between WC (cm) and height (cm), and the values ≥0.5 were considered as an indicator of fatness [29]. The conicity index (C-index), an index based on the similarity between accumulated fat around the waist and a cone that rises the metabolic risk, was obtained following the equation proposed by Valdez et al. [30]: WC (m)/[0.109 × √weight (kg)/height (m)]. 

Fat-free mass and fat mass (kg and percentage) were assessed by dual-energy X-ray absorptiometry (DXA; GE© Lunar densitometer, DPX NTVR, with ENCORE 2011 software, version 13.60, GE Healthcare, Chicago, IL, USA) in Goiania; by a tetrapolar single-frequency bioelectrical impedance analyzer (BIA) Biodynamic^®^ Model 450 (TBW, São Paulo, Brazil) in São Paulo and Fortaleza; and by BIA Quantum (RJL Systems™, Clinton Twp, MI, USA) in Pelotas. FMI and FFMI were calculated as fat mass (kg) and fat-free mass (kg) divided by height squared, respectively [31]. Although different BIA technologies with different algorithms were used to estimate body compartments, only the difference between the two assessments (pre-CT and post-CT) was considered for statistical analysis.

### 2.7. Statistical Analysis

The sample size was calculated based on the article published by Godinho-Mota et. al. [8], which has a similarity with the study aims. For the sample size, G*Power software version 3.1.9.2 was used, taking into consideration the CT effect on total body fat percentage [12]. The effect size of 0.575 showed that, with a significance level of 95% and statistical power of 80% (power 1-β 0.80), the minimum number of participants required was 42 patients. Thus, each region was included if it had at least 42 patients.

Data were processed using Excel software 10.0 (version 2013, Microsoft Corporation, Redmond, WA, USA), and statistical analysis was performed using SPSS software version 21.0 (IBM Corp., Chicago, IL, USA). Descriptive statistics were conducted to characterize the sample studied. Data presented in mean, standard deviation, and frequencies

The General Mixed Model (GMM) adjusted by age was used to verify the effect of CT regimens (pre-CT × post-CT) and the interaction between regimen and time on the variables investigated. Estimated marginal means and 95% confidence intervals (CI) were compared in pairs using Sidak for multiple tests. The level of significance for all analyses was set at *p* < 0.05.

## 3. Results

A total of 304 women with a mean age of 51 ± 11 years were evaluated in this study. Most women live with a partner (61.5%), receive from one to six minimum wages (42.8%), and are postmenopausal (40.1%) (Table 1). Most of the participants had invasive ductal carcinoma (71.4%), stage II (57.9%), luminal subtype (37.8%), and received the ACT regimen (48.7%) (Table 2). 

Most women are physically inactive (34.9%), do not report alcohol consumption (46.7%), and are non-smokers (54.9%) (Table 3). 

Considering all four regions studied, 62.5% (n = 190) of the patients pre-CT are overweight, and 65.9% (n = 195) post-CT. While 28.6% (n = 87) have a very high risk of metabolic complications, this increases to 30.1% (n = 89) after CT-treatment. The excess abdominal fat measured by WHtR (≥0.5) is 49.7% (n = 151) among patients before CT and it drops to 48.6% (n = 144) post-CT.

Although the anthropometric measurements of central obesity represented by WC, WHtR, and C-index do not present significative changes, a significant increase in BMI (∆ = 0.50 ± 0.09; *p* < 0.001) and FMI (∆ = 0.36 ± 0.10; *p* = 0.007) can be observed post-CT, independently of the regimen used (Table 4).

Overall, CT regimens have a significant increase in FMI and FFMI. Through the post-hoc Sidak test, the effects of the interaction between time and regimen can be found. The results indicate that, for the increase in BMI after CT, time seems to be the factor that has the most influence on this change (*p* < 0.001); for the increase in FMI after CT, both time (*p* = 0.007) and type of regime (*p* = 0.040) seem to influence this parameter of body composition, presenting significance for the ACT regimen (∆ = 0.56 ± 0.15). Finally, an increase in FFMI is probably more related to the type of CT regimen employed than the duration of the treatment itself, presenting significance for the AC regimen (∆ = 0.34 ± 0.12; *p* < 0.001) (Table 4).

## 4. Discussion

This study found that CT has an effect on BMI and FMI, as both increased after treatment. Regarding CT regimes, FMI increased significantly in women after ACT treatment, while FFMI increased after AC. These findings confirm our initial hypothesis that anthropometric and body composition parameters change throughout CT, and that the impact is different according to the type of CT regime.

Patients evaluated in this study were mainly classified as overweight in all centers. In the actual Brazilian epidemiologic scenario, the proportion of overweight women (+18 years) is ranged from 51.3% to 62.7% [32], a status that is positively associated with breast cancer risk [33,34]. Additionally, Brazilian breast cancer women presented higher BMI post-CT. The literature suggests that weight gain post treatment exists [8,9]; however, a recent cohort of Denmark women did not find an association between weight gained pre and post chemotherapy, even finding an average increase in body weight of 1.2 kg (*p* = 0.29) [35]. Although no significance was found in relation to the regimen used in our study, a meta-analysis [10] showed an increase of 2.7 kg (95% CI, 2.0–7.5) during CT, especially in women treated with CMF, where the weight gain was 3.5 kg (95% CI, 2.7–4.3) versus 1.4 kg (95% CI, 0.7–2.0) for non-CMF. Evidence suggests that CT regimens, mainly CMF, might conduct in greater body weight gain (8–10 kg), visceral adiposity, and reduction in FFMI regardless of age, energy food consumption, and clinical staging [10,13]. These changes can be driven by hormonal changes, given that estrogen suppression and insulin resistance induction can increase total fat and decrease fat-free mass, including skeletal muscle [36,37,38].

The central adiposity did not differ post-CT in this study. We used WC and WHtR measurements, which are highly used in population-based studies due to the viability of evaluation; but they can be imprecise indicators of intra-abdominal adipose tissue, as well subcutaneous fat deposition and visceral adipose tissue [39]. Computed tomography and magnetic resonance imaging are more accurate methods for assessing abdominal fat in breast cancer patients [40]; however, those methods have a higher cost and are not routinely used to assess breast cancer patients’ body composition in Brazil. 

In this current study, FMI increased in women post-CT with ACT regimen. The literature shows different findings related to CT regimens used by breast cancer patients on body composition [8,9,10]; however, ACT and CMF regimens are normally longer and can impact body composition [41]. Some women had a fat mass gain, possibly due to the abrupt hormonal changes and the ovarian collapse induced by the CT, which may increase fat mass in the visceral region and reduce bone mineral density [41].

It can be noticed that CT Impacts woman’s life independently of the regimen used (AC, ACT, CMF), affecting BMI and FMI values. In addition, CT can promote changes in physical activity and basal metabolic rate, which may be linked to changes in body composition [38]. During the treatment, breast cancer patients face daily routine changes starting with the chemo-infusion, which can be done in an outpatient setting or, depending on the case, during hospitalization [4]. Food intake and physical activity can also be changed due to side effects such as fatigue, reduced immunity, and severe anemia [42], which lead to less energy flow and consequent slowdown in metabolism [36,38]. For breast cancer patients, results reinforce the need to assess body components, as well as to consider the body fat disposition [8] and the presence of sarcopenia [42].

Interestingly, our study found an increase in FFMI in women who were using the AC regime through the post-hoc Sidak test. On the other hand, Del Rio et al. [43] found a significant increase in fat-free mass after the CMF regimen (43.6  ±  1.3 vs 45.2  ±  1.5 kg, *p*  < 0.001), which could be explained by the fluid retentions during the treatment that are conducive to a body water increase, and not a higher free-fat mass [19,44].

A potential limitation of our study was the use of non-standardized methods to assess body composition. Another limitation would be the lack of specific variables not collected in some centers (e.g., WC and WHtR), which resulted in a smaller sample size in certain analyses. Our study has some strengths. Firstly, the patient was included as a random variable in the GMM and the maximum time to complete CT (6 months) has been standardized. Secondly, the originality of the study and the involvement of different Brazilian regions allows a broader knowledge of the sociodemographic, clinical, hormonal, and therapeutic characteristics of Brazilian patients with breast cancer. Finding that there is an effect of weight gain and body composition during chemotherapy treatment and the impact of the regimes used was also key, highlighting that management weight recommendations should be considered during this treatment phase.

## 5. Conclusions

Our study concluded that there was an increase in BMI and FMI after CT, and the impact on FMI and FFMI was dependent on the CT regimen. FMI significantly increased in women after ACT treatment, while FFMI increased after AC. These results are important to disseminate to the scientific community and health professionals in order to adjust diet and physical activity to counterbalance body composition changes caused by CT treatment in breast cancer patients.

## Figures and Tables

**Figure 1 nutrients-15-01689-f001:**
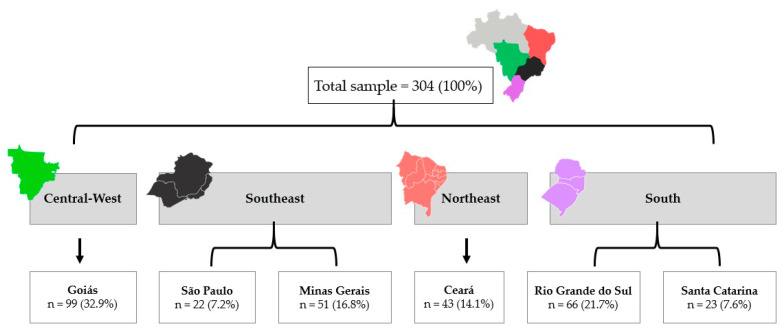
Women from six states of four regions of Brazil were included in this study.

**Table 1 nutrients-15-01689-t001:** Socioeconomic status characteristics of breast cancer women undergoing CT (*n* = 304), Brazil, 2004–2018.

Variables	*n* (%)
Socioeconomic status	
Age (Mean ± SD)	51.30 ± 10.77
Skin Color	
White	162 (53.3)
Nonwhite	142 (46.7)
Marital status	
With partner	187 (61.5)
Without partner	117 (38.5)
Level of education attained	
Grade school incomplete	65 (21.4)
Grade school complete	16 (5.3)
High school incomplete	18 (5.9)
High school complete	62 (20.4)
Undergraduate degree incomplete	17 (5.6)
Undergraduate degree complete	30 (9.9)
Unlettered	10 (3.3)
NR	86 (28.3)
Family income/month by minimum wage	
<1	54 (17.8)
>1 to ≤6	130 (42.8)
>6 to <10	16 (5.3)
≥10	8 (2.6)
0	3 (1.0)
NR	93 (30.6)
Menopausal status	
Pre-menopausal	89 (29.3)
Postmenopausal	122 (40.1)
NR	93 (30.6)

NR, Not reported.

**Table 2 nutrients-15-01689-t002:** Tumor characteristics of breast cancer women undergoing CT (*n* = 304), Brazil, 2004–2018.

Variables	*n* (%)
Morphological types	
Ductal	217 (71.4)
Lobular	11 (3.6)
Special type (Mucinous)	3 (1.0)
NR	73 (24.0)
TNM Clinical stage	
I	42 (13.8)
II	176 (57.9)
III	86 (28.3)
Grade	
G1	39 (12.8)
G2	97 (31.9)
G3	62 (20.4)
Gx	4 (1.3)
NR	102 (33.6)
Estrogen receptor	
Positive	139 (45.7)
Negative	55 (18.1)
NR	110 (36.2)
Progesterone receptor	
Positive	123 (40.5)
Negative	71 (23.4)
NR	110 (36.2)
HER2	
Positive	38 (12.5)
Negative	147 (48.4)
NR	119 (39.1)
Molecular subtypes	
Luminal	115 (37.8)
HER2	29 (9.5)
Triple Negative	38 (12.5)
NR	122 (40.1)
CT regimen	
AC	91 (29.9)
ACT	148 (48.7)
CMF	35 (11.5)
PD	13 (4.3)
NR	17 (5.6)
Surgery type	
No surgery	22 (7.2)
Mastectomy	90 (29.6)
Breast-conserving surgery	102 (33.6)
NR	90 (29.6)

AC, anthracyclines; ACT, anthracyclines and taxane; CMF, cyclophosphamide, methotrexate, luorouracil; PD, isolated paclitaxel/docetaxel; G1, grade 1 tumor (well-differentiated); G2, grade 2 tumor (moderately differentiated); G3, grade 3 tumor (poorly differentiated); Gx, the grade cannot be assessed; HER2, human epidermal growth factor receptor-type 2; TMN, tumor node, and metastasis; NR, Not reported.

**Table 3 nutrients-15-01689-t003:** Behavioral variables of breast cancer women undergoing CT (*n* = 304), Brazil, 2004–2018.

Variables	*n* (%)
Alcohol consumption (grams of ethanol/week)	
≥10	49 (16.1)
<10	142 (46.7)
NR	113 (37.2)
Smoking status	
Current smokers	25 (8.2)
Non-smokers	167 (54.9)
Ex-smokers	65 (21.4)
NR	47 (15.5)
Physical activity (minutes/week)	
Active (≥150)	67 (22.0)
Inactive (<150)	106 (34.9)
NR	131 (43.1)

NR, Not reported.

**Table 4 nutrients-15-01689-t004:** Effect of the CT protocol and time on anthropometry and body composition, Brazil, 2004–2018.

		CT Regimen				Interaction *	
Dependent Variables	Overall Effect	AC	ACT	CMF	PD	Effects	*p*
BMI (n = 287)		28.65 ± 0.56	28.36 ± 0.44	27.85 ± 0.94	27.93 ± 1.48	Time	<0.001
Pre-CT	27.92 ± 0.48 ^a^	28.34 ± 0.56	28.14 ± 0.45	27.63 ± 0.95	27.56 ± 1.49	Regimen	0.885
Post-CT	28.47 ± 0.48 ^b^	28.96 ± 0.57	28.58 ± 0.45	28.06 ± 0.95	28.29 ± 1.49	Time * Regimen	0.802
∆	0.50 ± 0.09	0.62 ± 0.13	0.45 ± 0.15	0.43 ± 0.18	0.71 ± 0.50		
WC * (n = 173)		89.48 ± 2.54	91.61 ± 1.11	84.31 ± 5.17	92.16 ± 3.82	Time	0.248
Pre-CT	88.92 ± 1.78	89.24 ± 2.57	91.05 ± 1.13	83.55 ± 5.25	91.84 ± 3.87	Regimen	0.495
Post-CT	89.86 ± 1.82	89.72 ± 2.64	92.16 ± 1.16	85.08 ± 5.37	92.49 ± 3.97	Time * Regimen	0.956
∆	1.02 ± 0.46	0.48 ± 0.91	1.14 ± 0.58	1.53 ± 1.70	0.65 ± 1.60		
WHtR * (n = 173)		0.56 ± 0.02	0.58 ± 0.01	0.55 ± 0.04	0.59 ± 0.03	Time	0.950
Pre-CT	0.57 ± 0.01	0.56 ± 0.02	0.58 ± 0.01	0.55 ± 0.04	0.59 ± 0.03	Regimen	0.554
Post-CT	0.57 ± 0.01	0.57 ± 0.02	0.59 ± 0.01	0.55 ± 0.04	0.58 ± 0.03	Time * Regimen	0.869
∆	0.00 ± 0.00	0.01 ± 0.01	0.00 ± 0.00	0.00 ± 0.00	0.01 ± 0.01		
C-index * (n = 173)		1.25 ± 0.02	1.26 ± 0.01	1.24 ± 0.04	1.25 ± 0.03	Time	0.730
Pre-CT	1.25 ± 0.13	1.25 ± 0.02	1.26 ± 0.01	1.22 ± 0.04	1.26 ± 0.03	Regimen	0.807
Post-CT	1.25 ± 0.13	1.24 ± 0.02	1.26 ± 0.01	1.25 ± 0.04	1.25 ± 0.03	Time * Regimen	0.671
∆	0.00 ± 0.00	−0.01 ± 0.01	0.00 ± 0.01	0.03 ± 0.02	−0.01 ± 0.02		
FMI ^†^ (n = 213)		10.54 ± 0.44	11.76 ± 0.35	9.97 ± 0.71	8.98 ± 2.12	Time	0.007
Pre-CT	10.02 ± 0.59 ^a^	10.44 ± 0.45	11.48 ± 0.36 ^a^	9.90 ± 0.72	8.27 ± 2.16	Regimen	0.040
Post-CT	10.60 ± 0.59 ^b^	10.63 ± 0.45	12.04 ± 0.36 ^b^	10.05 ± 0.72	9.70 ± 2.16	Time * Regimen	0.121
∆	0.36 ± 0.10	0.17 ± 0.14	0.56 ± 0.15	0.15 ± 0.13	1.44 ± 2.24		
FFMI ^†^ (n = 213)		18.04 ± 0.28 ^a^	14.87 ± 0.22 ^b^	18.21 ± 0.45 ^a^	16.40 ± 1.34 ^a,b^	Time	0.405
Pre-CT	16.82 ± 0.37	17.87 ± 0.28 ^a^	14.89 ± 0.23	18.05 ± 0.46	16.46 ± 1.36	Regimen	<0.001
Post-CT	16.95 ± 0.37	18.21 ± 0.29 ^b^	14.86 ± 0.23	18.38 ± 0.46	16.34 ± 1.36	Time * Regimen	0.06
∆	0.15 ± 0.07	0.34 ± 0.12	−0.04 ± 0.10	0.34 ± 0.16	−0.12 ± 0.12		

AC, anthracyclines; ACT, anthracyclines and taxane; BMI, body mass index; C-index, conicity index; CMF, cyclophosphamide, methotrexate, and 5-fluoracil; FMI, Fat mass index; FFMI, fat-free mass index; PD, isolated paclitaxel/docetaxel; T1, post-diagnosis; T2, after last CT cycle; WC, waist circumference, W/H ratio, waist circumference and height ratio; ∆: variation post-pre chemotherapy. Data presented as mean ± standard error. BMI, W/H ratio, C-index, FMI, FFMI: AR1 matrix for repeated effect and matrix of variance components for random effect. General Mixed Models: Data adjusted for age; WC: Diagonal matrix for repeated effect and matrix of variance components for random effect. ^ab^ Represent statistical significance in comparison of pairs through the Post-hoc Sidak analysis; *p*-value < 0.05. * Analysis considering the GO, MG, and SC states. ^†^ Analysis considering the GO, RS, SP, and CE states.

## Data Availability

The data presented in this study are available on request from the corresponding author.

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
