# Peer review of "Impact of Chemotherapy Regimens on Body Composition of Breast Cancer Women: A Multicenter Study across Four Brazilian Regions"

_nutrients, 2023, doi:10.3390/nu15071689_

Round 1

Reviewer 1 Report

In this manuscript, the authors describe about the influence of therapeutic treatment to body compositions in Brazilian breast cancer patients. BMI and FMI are increased after chemotherapy, and FMI and FFMI are also affected by chemotherapy. The authors suggest that nutritional intervention and physical activity programs could be developed and encouraged to counterbalance the changes in body composition caused by chemotherapy in breast cancer patients. Although this manuscript is somehow interesting, there are concerns to publish at this time.

Comments

1. I could not understand the significance of the results. The result sections just show the statistical data and there is no further consideration. For example, which is the most important factor associated with the increase of BMI/FMI after chemotherapy? Is chemotherapy directly affects to the body compositions?

2. The authors should divide or modify the table so that the readers can easily find the item.

Author Response

Dear editors and reviewers,

We appreciate the thoughtful reviews provided and their recommendations for improvement. Please find below a point-by-point response to the reviewer’s comments.

Warm Regards,

Jordana Godinho-Mota

Reviewer 1

I could not understand the significance of the results. The result sections just show the statistical data and there is no further consideration. For example, which is the most important factor associated with the increase of BMI/FMI after chemotherapy? Is chemotherapy directly affects to the body compositions? 

Thank you for your comment. Although we have not found significant changes in the central obesity represented by WC, WHtR, and C-index, a significant increase in BMI and FMI were observed after CT treatment regardless of the CT regimen used. This statement was added in lines 214-217. Chemotherapy does directly affect body composition. Through the results, we found an increase in FMI post-CT for the ACT regimen and an increase in FFMI post-CT for the AC regimen. We agree that this information was not clear, and we changed the writing to be clear and focused on the findings (please, see lines 218-225).

The authors should divide or modify the table so that the readers can easily find the item.

Thank you for the suggestion. We have divided the tables presented to make the results clearer. Also, we rewrote the results focusing on the aim of the manuscript.

Reviewer 2 Report

This interesting article reports changes in BMI including fat free mass index and fat mass index, as a result of chemotherapy treatment in 304 women with breast cancer. This manuscript is well written and conveys the information clearly. 

The introduction could introduce the socioeconomic analysis of the cohort, as this information is also interesting to a general audience.

Are there significant differences in table 1? P-values should be reported for all significant differences between groups/regions. The result section between tables could be expanded to include this information.

Author Response

March 15 th, 2023

Dear editors and reviewers,

We appreciate the thoughtful reviews provided and their recommendations for improvement. Please find below a point-by-point response to the reviewer’s comments.

Warm Regards,

Jordana Godinho-Mota

Reviewer 2

The introduction could introduce the Introducing the socioeconomic analysis of the cohort, as this information is also interesting to a general audience.

We appreciate your suggestion and agree with it. We added this reflection, which can be seen in lines 43–52.

Are there significant differences in table 1? P-values should be reported for all significant differences between groups/regions. The result section between tables could be expanded to include this information.

Thank you very much for your observation. We included a representative flowchart (Figure 1) of the number of women in each region/state included in the study eliminating tables 1 and 2 by region. We kept only the data related to the total sample which was our main objective.

Author Response

March 15 th, 2023

Dear editors and reviewers,

We appreciate the thoughtful reviews provided and their recommendations for improvement. Please find below a point-by-point response to the reviewer’s comments.

Warm Regards,

Jordana Godinho-Mota

Reviewer 3

The strongest part of this study is the methodology, the anthropometric measurement and statistical calculations. However,  The section of results can be improved. Please highlight the main results of this study.

Thank you for your comment. We have improved the results section highlighting our main results. Please refer to lines 187-245.

The discussion section must be improved. It is unclear. Line 269 The authors must clarify why the CT therapy increase the BMI and visceral adiposity ? Please identify possible mechanisms!!!

Thank you for your comment. We provided mechanisms that clarified why CT therapy increases BMI and visceral adiposity (Please see lines 265-267).

Apparently, the CT treatment increases the BMI, as stated by the authors as a negative effect. Line 257, 286, 287 I consider that negative effect is an inappropriate statement. It should be written an increase in BMI, etc

Line 281 …it is unclear which of the regimen can have the worst impact on body composition .. I consider again an inappropriate statement.

Thank you for the suggestion. We have rewritten the sentences.  

Line 287 The authors stated that CT therapy intensify changes in physical activity and basal metabolic rate? How ? Please clarify. Intensify is an inappropriate term used here.

Thanks for your comment. We have added new references that clarify the link between chemotherapy and changes in physical activity and food intake experienced by patients during CT therapy (please see lines 285-289).

Round 2

Reviewer 1 Report

The authors carefully responded to my comments and now I think this manuscript is acceptable for publication.